# Phylogeny and species delimitation of the genus *Longgenacris* and *Fruhstorferiola viridifemorata* species group (Orthoptera: Acrididae: Melanoplinae) based on molecular evidence

Jingxiao Gu[1,2�], Bing Jiang[1,2�], Haojie Wang[3], Tao Wei[4], Liliang Lin[5], Yuan Huang[5], Jianhua Huang[1,2]*

**1** Key Laboratory of Insect Evolution and Pest Management for Higher Education in Hunan Province, Central South University of Forestry and Technology, Changsha, Hunan, People's Republic of China, **2** Key Laboratory of Cultivation and Protection for Non–Wood Forest Trees (Central South University of Forestry and Technology), Ministry of Education, Changsha, Hunan, People's Republic of China, **3** Center for Computational Biology, College of Biological Sciences and Technology, Beijing Forestry University, Beijing, People's Republic of China, **4** Tanxi Street Agency, Liunan Subdistrict, Liuzhou, Guangxi, People's Republic of China, **5** College of Life Sciences, Shaanxi Normal University, Xi'an, Shaanxi, People's Republic of China

☺ These authors contributed equally to this work.
* caniscn@aliyun.com

**Data Availability Statement:** All new haplotype nucleotide sequences are available from the

## Abstract

Phylogenetic positions of the genus *Longgenacris* and one of its members, i.e. *L. rufiantennus* are controversial. The species boundaries within both of *L. rufiantennus+Fruhstorferiola tonkinensis* and *F. viridifemorata* species groups are unclear. In this study, we explored the phylogenetic positions of the genus *Longgenacris* and the species *L. rufiantennus* and the relationships among *F. viridifemorata* group based on the 658-base fragment of the mitochondrial gene cytochrome c oxidase subunit I (*COI*) barcode and the complete sequences of the internal transcribed spacer regions (*ITS1* and *ITS2*) of the nuclear ribosomal DNA. The phylogenies were reconstructed in maximum likelihood framework using IQ-TREE. K2P distances were used to assess the overlap range between intraspecific variation and interspecific divergence. Phylogenetic species concept and NJ tree, K2P distance, the statistical parsimony network as well as the generalized mixed Yule coalescent model (GMYC) were employed to delimitate the species boundaries in *L. rufiantennus+F. tonkinensis* and *F. viridifemorata* species groups. The results demonstrated that the genus *Longgenacris* should be placed in the subfamily Melanoplinae but not Catantopinae, and *L. rufiantennus* should be a member of the genus *Fruhstorferiola* but not *Longgenacris*. Species boundary delimitation confirmed the presence of oversplitting in *L. rufiantennus+F. tonkinensis* and *F. viridifemorata* species groups and suggested that each group should be treated as a single species.

GenBank database (accession numbers are given in S2 Table).

**Funding:** This study is supported by the Open foundation for innovation platform of Education Bureau of Hunan Province (18K056) and National Natural Science Foundation of China (No. 31540055, 31260523, 31801993). The funders had no role in study design, data collection and analysis, decision to publish, or preparation of the manuscript.

**Competing interests:** The authors have declared that no competing interests exist.

## Introduction

Taxonomy is a process to take or collate decisions continually. Any taxonomic decision taken since the inception of zoological nomenclature in 1758 has relevance today, and on into the future, no matter that decision was right or wrong [1]. The process of modern taxonomy can be viewed as a taxonomic circle, and hypothesis established from any information should be tested with other sources of information, i.e. taxonomists must break out of the circle of inference in species delineation work to raise the entity to species status [2].

Cryptic species usually refers to as one of two or more species that are morphologically indistinguishable in adult stage and incapable of interbreeding since most morphospecies were described based on adult types so far. Cryptic species have been detected in some insect groups through molecular evidences and tested with other information such as morphological, geographical, biological, ecological and DNA sequence evidences [3–11]. It is clear that genomic information should be an active component of modern taxonomy, and that integration of the "fashionable" molecular approaches with the classical taxonomic approach is a critical component of reconciling both camps [2, 12, 13].

Despite the existence of over lumping (cryptic species), oversplitting may also exist especially in early described species groups because of the lack of type comparison, which usually lead to repeated descriptions of the same species as different ones without actual morphological difference [14]. Incorrect assignment of a species in genus or higher levels will also lead to description of the same species as different ones because the comparison can't be made between the most close relatives. In these cases, morphological revision is necessary to confirm the presence of morphological differences among the closely related species. Moreover, other sources of data should be used to determine species boundary and test species hypotheses [2].

*Longgenacris* is a grasshopper genus belonging to subfamily Melanoplinae with *L. maculacarina* You & Li, 1983 as type species [15–17]. The second species of the genus, *L. rufiantennus* Zheng & Wei, 2003, was described based on materials from Xiaolong, Yizhou, Guangxi, China [18], but recently transferred to the genus *Fruhstorferiola* Willemse, 1922 and synonymized with *F. tonkinensis* (Willemse, 1921) based on morphological similarity [16].

*Fruhstorferiola* is also a genus in Melanoplinae with 13 known species worldwide [17]. According to the shape of cercus, *Fruhstorferiola* can be tentatively divided into three species groups: (1) *F. viridifemorata* group, with cercus of male laterally compressed and expanded into boot-shaped apically (Fig 1A), (2) *F. tonkinensis* group, with cercus of male not expanded into boot-shaped apically but slender and slightly spear-shaped (Fig 1B), and (3) *F. huangshanensis* group, with cercus of male laterally compressed and semiroundly expanded in apical half but not boot-shaped (Fig 1C). Among the 13 known *Fruhstorferiola* species, 7 species distributed in continental China belong to *F. viridifemorata* group, with 4 species, i.e. *F. viridifemorata*, *F. kulinga*, *F. huayinensis* and *F. omei*, widespread and the remaining 3 species, i.e.

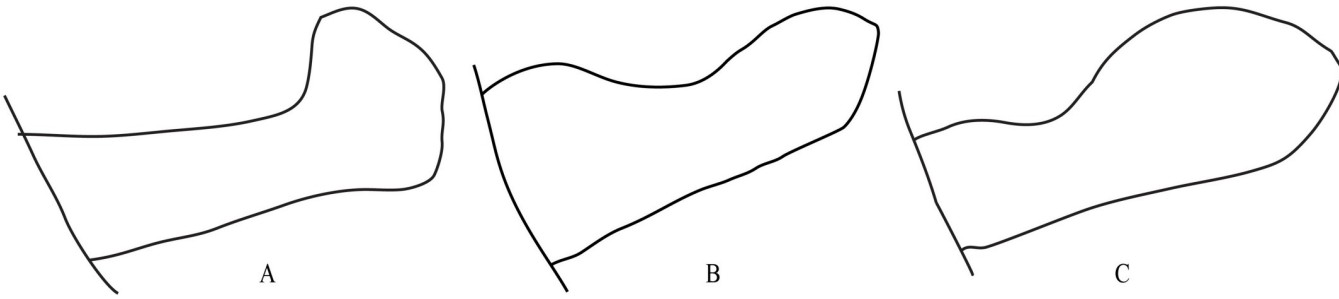

**Fig 1. Shape of male cerci in *Fruhstoferiola* spp.** A. *F. viridifemorata*. B. *F. tonkinensis*. C. *F. huangshanensis*.

*F. brachyptera*, *F. rufucorna* and *F. xuefengshana*, having been recorded only from the type locality. The main morphological characters used to distinguish species in *F. viridifemorata* group from each other are the length of tegmen, the shape of male cercus in apical portion and teeth in the posterior margin of female subgenital plate. However, these characters vary even among individuals from the same population. For example, specimens of each species collected from the same locality on the same date exhibit similar pattern of variation in tooth length (Fig 2), with median tooth longer than submedian and lateral teeth in some individuals (Fig 2A, 2C, 2E and 2G), but nearly as long as (Fig 2B, 2D, 2F and 2H) or slightly shorter (Fig 2I) than submedian and lateral teeth in other individuals, or with submedian teeth indistinct or even absent in a few individuals (Fig 2J). Therefore, it is difficult to identify specimens of *F. viridifemorata* group using morphological characters only, and frequently the same specimen could be probably recognized as different species by different identifiers.

Species delimitation using molecular data has attracted more and more attention from systematists and taxonomists because of the rapid development of sequencing techniques and bioinformatic methods. There are many successful cases in grasshoppers using molecular evidence for species delimitation so far [5, 6, 8, 9, 11–14, 19–21]. Molecular approaches for

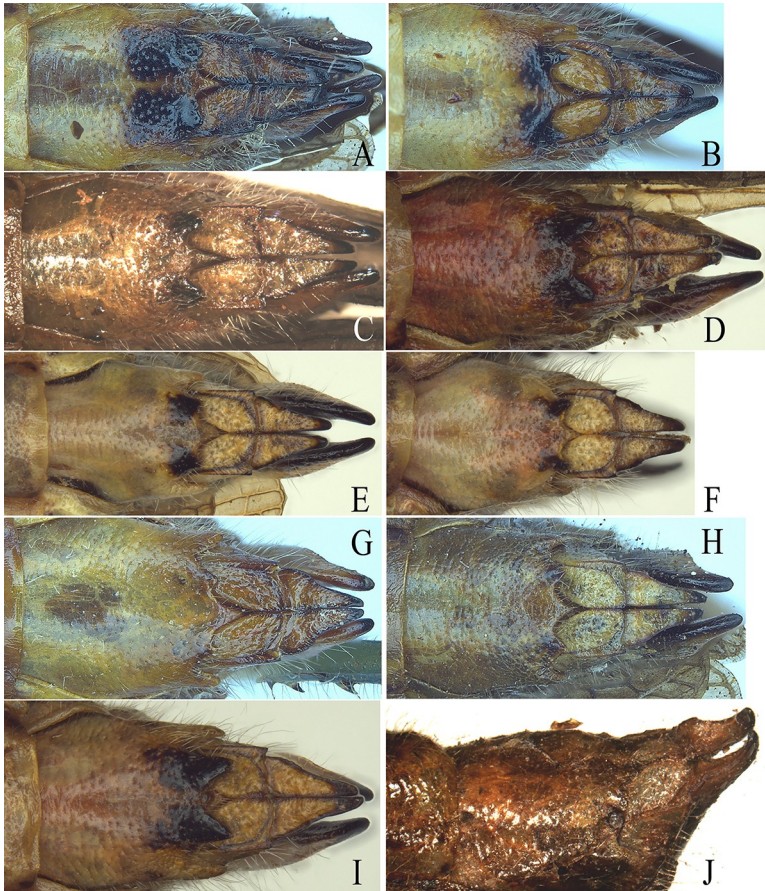

**Fig 2. Variation of teeth in posterior margins of female subgenital plates of *Fruhstorferiola* spp.** A-B, J. *F. viridifemorata*. C-D, I. *F. omei*. E-F.*F. kulinga*. G-H. *F.huayinensis*. A, C, E, G. The condition with median tooth distinctly longer than submedian and lateral teeth. B, D, F, H. The condition with median tooth nearly as long as submedian and lateral teeth. I. The condition with median tooth slightly shorter than submedian and lateral teeth. J. The condition with submedian teeth absent.

species delimitation can be used not only to confirm delimitations proposed by traditional taxonomy [11], but also to delimit new species under an integrative taxonomy framework despite the possibility of oversplitting sometimes [9]. As for the molecular markers, the most frequently employed one is the mitochondrial gene *COI*, which was used either alone [9, 14] or together with some other markers [5, 8, 11, 20].

To clarify the phylogenetic position of *L. rufiantennus* and the relationships among *F. viridifemorata* group, we sequenced the 658-base fragment of the 5' end of *COI* corresponding to the barcode region for animals [22], and the complete sequences of *ITS1* and *ITS2* of the nuclear ribosomal DNA from149 individuals belonging to 7 genera and 12 species in Acrididae, 1 individual in Tetrigidae and 2 individuals in Tettigoniidae. The phylogeny of the species involved was reconstructed from molecular sequence dataset using maximum likelihood method, and the species boundary was delimited using multiple methods, including genetic distance, NJ tree, the haplotype network constructed using the statistical parsimony method [23], and analysis of the generalized mixed Yule coalescent model (GMYC) [24].

## Materials and methods

### Taxon sampling

A total of 152 individuals representing 3 families 9 genera and 14 species were sampled (S1 Table). At least five individuals from each population and as many populations as possible of the widespread species were sampled whenever the specimens were available (S1 Table). Species assignation of specimens was performed mainly following Li & Xia's [25] key to species plus geographical information. For example, the specimens from type locality and neighboring places will be assigned to the same species if there is no distinct difference between them. Partial *COI* sequences were from our previous study (S2 Table) [14]. All specimens were preserved in anhydrous ethanol and stored at room temperature.

### DNA extraction, PCR amplification and sequencing

Whole genomic DNA was extracted from muscle tissue of the hind femur using a routine phenol/chloroform method [26]. The primers for the amplification of *COI* fragment followed our previous study: COBU (5'-TYTCAACAAAYCAYAARGATATTGG-3') and COBL (5'-TAAACTTCWGGRTGWCCAAARAATCA-3') [14]. Amplification of complete sequences of *ITS1* and *ITS2*employed the following primers: 18sF1 (5'-ATGTGCGTTCRAAATGTCGATGTTCA-3') and 5.8sB1d (5'-ATGTGCGTTCRAAATGTCGATGTTCA-3') for *ITS1* [27], ITS3 (5'-GCATCGATGAAGAACGCAGC-3') and ITS4 (5'-TCCTCCGCTTATTGATATGC-3') for *ITS2* [28].

PCRs were carried in a 25 μL reaction mixture containing 13.875 μL of ultrapure water, 2.5 μL of 10×PCR buffer (Mg2+free), 2.5 μL of MgCl$_2$ (25 mM), 2 μL of dNTP (2.5 mM), 1.5μL of each primer (0.01 mM), 0.125 μL of TaKaRa r-Taq polymerase, and 1 μL of DNA template. The cycling protocol consisted of an initial denaturation step at 95°C for 5 min, followed by 30–35 cycles of denaturation at 94°C for 45 s, annealing at 48°C for 45 s and extension at 70°C for 1 min 30 s, and a final extension at 72°C for10 min and then held at 4°C. PCR products were sent to Sangon Biotech (Shanghai) Co., Ltd and sequenced bidirectionally after purification. Sequencing primers were the same as those for PCR amplification.

### Sequence assembly and alignment

Assembly of the raw sequencing files was implemented in the Staden Package [29]. The assembled sequences were aligned using Clustal X [30], and the primer sequences in both ends of the

sequences were excised to remove artificial nucleotide similarity derived from PCR amplification. *COI* nucleotide sequences were translated into amino acid sequences to detect the potential nuclear mitochondrial pseudogenes (numts) based on the presence of premature stop codons and shifts in reading frame [31, 32]. Haplotype nucleotide sequences were deposited in GenBank (MH934098—MH934186, S2 Table). Each haplotype was blasted using MEGA-BLAST option against the nucleotide collection (nr/nt) available on the NCBI website (http://blast.ncbi.nlm.nih.gov/Blast.cgi?CMD=Web&PAGE_TYPE=BlastHome). Only haplotypes that blasted within the correct suborder with E-values $\leq$ 1.00E-30 were included in this study [33]. The combined data set of *COI*, *ITS1* and *ITS2* was concatenated in SequenceMatrix [34].

## Intraspecific variation, interspecific divergence and phylogeny reconstruction

Sequence divergences were calculated using the Kimura two parameter (K2P) distance model [35, 36]. The calculation of the sequence divergences was implemented in MEGA7.0 [37].

The phylogenies were reconstructed in maximum likelihood framework with *Ergatettix dorsiferus* in Tetrigidae and *Conocephalus longipennis* in Tettigoniidae as outgroups. Maximum-Likelihood phylogenies were reconstructed using IQ-TREE [38], a fast and effective stochastic algorithm combining hill-climbing approaches and a stochastic perturbation method, best-fit models of nucleotide evolution and best-fit partitioning scheme were selected using ModelFinder [39], the approximately unbiased branch support values were calculated using UFBoot2 [40], and the analysis was performed in W-IQ-TREE [41] using default sets most of the time.

To provide a profile for the setup of taxa and groups for calculating genetic distances, a neighbor-joining (NJ) tree of K2P distances was created to provide a graphic representation of the patterning of divergence between species [42] because of its strong track record in the analysis of large species assemblages [43]. NJtree building with 1000 bootstrap replicates was implemented inMEGA7 [37].

## Network analysis

Statistical parsimony network [23] can provide more significant inferences about evolutionary relationships than traditional bifurcating trees when divergences are low. The 95% parsimony connection limit may be used as an objective standard of genetic differentiation for the identity of traditional species or evolutionarily significant units (ESUs) [44, 45]. In most of published network analyses, alignments of DNA sequences typically fall apart into a separate subnetwork for each Linnean species (but with a higher rate of true positives for mtDNA data) and DNA sequences from single species typically stick together in a single haplotype network [46]. Therefore, we constructed haplotype networks for *Longgenacris* species and *F. viridifemorata* group. The construction of haplotype networks was implemented in TCS1.21 [47].

## Analysis of the generalized mixed Yule coalescent model (GMYC)

The single-threshold GMYC analyses were conducted in R v3.6.1 in a Windows environment with the use of the *splits* package. The ultrametric single-locus gene tree required for the GMYC method was obtained using BEAST 1.8.2 [48] with 10 million MCMC generations under the Yule speciation model. A strict molecular clock was shown to be appropriate to infer the ultrametric trees through the model comparison using a Bayes factor test in Tracer 1.6. Effective sample sizes (ESS) and trace plots estimated with Tracer 1.6 were used as convergence diagnostics, and a burn-in of one million generations was used to avoid suboptimal trees in the final consensus tree.

## Results

### Phylogeny

Phylogeny of the taxa involved in this study was reconstructed in maximum likelihood framework using separate alignments of *COI*, *ITS1*, *ITS2* and their concatenated alignment, respectively.

The trees inferred from *COI* and the combined alignments displayed similar topologies (Fig 3). Nearly all species formed reciprocally monophyletic clades except *F. tonkinensis*+ *L. rufiantennus* and *F. viridifemorata* groups. The main differences between the single *COI* gene tree and the combined alignment tree were the placements of *Emeiacris maculata*, of which two clades did not form monophyletic clade but were added in turn to the clade of its closest relative *Paratonkinacris vittifemoralis* in *COI* gene tree (Fig 3A), and *Apalacris tonkinensis*, which is a member of the subfamily Catantopinae but had a closer relationship to most of Melanoplinae members than *Tonkinacris sinensis* in the combined alignment tree (Fig 3D), i.e. Melanoplinae formed a monophyletic clade in *COI* gene tree but not in the combined alignment tree. For the clade of *F. tonkinensis*+ *L. rufiantennus* group, all of the 15 individuals of *L. rufiantennus* scattered within the clade of *F. tonkinensis* (Fig 3B and 3E). Individuals within *F. viridifemorata* group clustered neither by species nor populations (Fig 3C and 3F). The four individuals of *F. kulinga* from Longmenhe, Xingshan County, Hubei Province exhibited most complicated relationship with other species/populations, with two individuals close to *F. omei* and *F. viridifemorata*, one close to *F. huayinensis*, and the remaining one close to individuals of *F. kulinga* from Hunan and Guangxi populations in tree from combined dataset or located at the base of the tree from *COI* gene (Fig 3C and 3F). *L. maculacarina* consistently formed a monophyletic clade and had a most close relationship to *Fruhstorferiola* species (Fig 3A and 3D).

The tree inferred from *ITS1* sequences had less resolution at species level (S1 Fig). Although *F. tonkinensis*+*L. rufiantennus* group formed a monophyletic clade, but it falled into the clade of *F. viridifemorata* group and all *Fruhstorferiola* species formed a larger monophyletic clade. *P. vittifemoralis* and *E. maculata*formed a large monophyletic clade together, but neither of them formed monophyletic subclade. The remaining four distantly related species formed monophyletic clades each, except one individual of *L. maculacarina* falled into the clade of *T. sinensis*. *Apalacris tonkinensis* falled into the members of the subfamily Melanoplinae just as in the combined alignment tree. The tree inferred from *ITS2* sequences had a similar topology to that from *ITS1* sequences, but all members of the tribe Melanoplinae formed a large monophyletic clade as in *COI* gene tree (S2 Fig).

As for NJ trees, the one deduced from single *COI* gene (S3A Fig) had extremely similar topology to that of ML tree. Monophyletic clades could be retrieved consistently for distantly related species and closely related species groups both in single and combined alignment trees with exceptions for only a few individuals in single *ITS1* and *ITS2* alignment trees. For example, individual gh016 of *L. maculacarina* falled into the clade of *T. sinensis* in the tree from *ITS1* sequences (S3B Fig), individual gh041 of *F. tonkinensis* falled into the clade of *O. longipennis* in the tree from *ITS2* sequences, individual gh086 of *F. omei*, gl0097 of *F. huayinensis*, gh075 of *E. maculata* and gh080 of *L. rufiantennus* escaped from their own stem clades, respectively (S3C Fig). Monophyly of Melanoplinae was supported in both single and combined alignment trees (S3A–S3D Fig).

### Intraspecific variation and interspecific divergence

Based on the neighbor-joining (NJ) tree of K2P distances, taxa or groups were set up to calculate the intraspecific variations and interspecific divergences. The results showed that, for *COI*

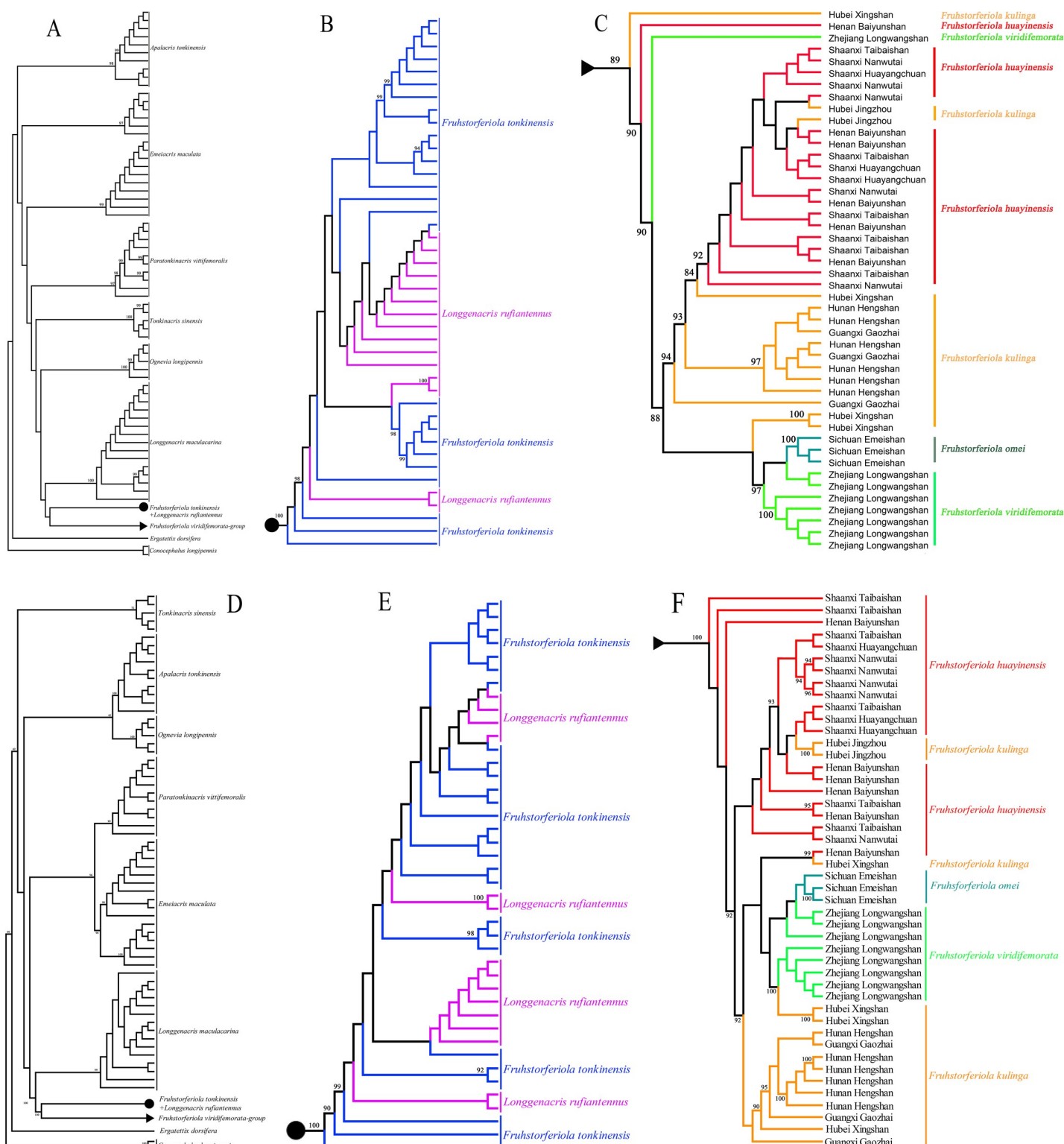

**Fig 3. Phylogeny deduced in maximum likelihood framework from alignment of *COI* gene and concatenated alignment of *COI* gene, *ITS1* and *ITS2* sequences.**
A–C. Cladogram deduced from *COI* gene. D–F. Cladogram deduced from concatenated alignment of *COI* gene, *ITS1* and *ITS2* sequences. A, D. Full trees with subclade of *L. rufiantennus*+*F. tonkinensis* group and *F. viridifemorata* group collapsed. B, E. Subclades of *L. rufiantennus*+*F. tonkinensis* group. C, F. Subclade of *F. viridifemorata* group.

sequences, variations within population were mostly distinctly less or slightly larger than 1%, except those of *F. kulinga* within Longmenhe population, of which the maximum pairwise distance was 2.33%; intraspecific variations between populations were usually less than 3% (S3 Table), a putative threshold for species assignment proposed by previous study (Herbert *et al*., 2003), with *E. maculata* as the single exception which had much higher intraspecific variations (4.24–4.73%, average 4.45%) between interpopulation individuals. Two populations of *E. maculata*, one from Hengshan of Hunan and the other from Emeishan of Sichuan, were sampled; the variations within population were less than 1% but those between populations ranged from 4.24% to 4.73%. Both *ITS1* and *ITS2* sequences showed much lower intraspecific variations but had similar distribution pattern (S3 Table).

The interspecific divergences of *COI* sequences within *F. viridifemorata* groups ranged from 1.00% to 2.03%, those between species of *F. viridifemorata* groups and *F. tonkinensis* were up to 5.53–6.08% and the one between *F. tonkinensis* and *L. rufiantennus* was 0.33%, but that between *L. rufiantennus* and *L. maculacarina* was as high as 7.33% (S4 Table). The interspecific divergences calculated from *ITS1* and *ITS2* sequences displayed similar distribution patterns (S5 and S6 Tables), i.e. species within *F. viridifemorata* group and *F. tonkinensis*+ *L. rufiantennus* group had much lower between-species mean distances but the mean distances between other pairwise species were distinctly much higher. For all of three alignments, the distances between species within Melanoplinae were constantly lower than those between species in Melanoplinae and that out of Melanoplinae (S4–S6 Tables).

## Speciesboundary delimitation

**(1)** ***Fruhstorferiola tonkinensis + Longgenacris rufiantennus* group.**   Considering the high similarity between *F. tonkinensis* and *L. rufiantennus*, we sampled 15 individuals of *L. rufiantennus* from its type locality, 27 individuals of *F. tonkinensis* in total from five populations and 15 individuals of *L. maculacarina* for comparison. The results showed that *L. maculacarina* usually formed a monophyletic clade, but all individuals of *L. rufiantennus* fell completely into the clade of *F. tonkinensis* in NJ trees reconstructed both from single and combined alignment sequences (Fig 4A, S3A–S3D Fig), with only one exceptive individual for each species escaping from its own stem clade in NJ tree of *ITS2* sequences, i.e. individual gh080 of *L. rufiantennus* clustered into a clade together with gh075 of *E. maculata* and individual gh016 of *L. maculacarina*, and individual gh041 of *F. tonkinensis* falled into the clade of *O. longipennis* (S3C Fig).

For *COI* sequences, mean intraspecific variations within each species were all distinctly less than 1% (Table 1). Pairwise intraspecific variations within *F. tonkinensis* ranged from 0 to 1.08%, and that within *L. rufiantennus* ranged from 0 to 0.46%. Pairwise interspecific divergence between *F. tonkinensis* and *L. rufiantennus* ranged from 0 to 0.77%, and completely fell into the range of pairwise intraspecific variations within *F. tonkinensis*. Pairwise interspecific divergence between *L. rufiantennus* and *L. maculacarina* ranged from 7.24–7.92% and the mean divergence was 7.33% (Table 1). For *ITS1* and *ITS2* sequences, both intraspecific variations and interspecific divergences were much lower but had similar variation patterns (S3, S5 and S6 Tables).

Analysis with haplotype network led to a similar result. The numbers of *COI* haplotypes detected in *F. tonkinensis*, *L. rufiantennus* and *L. maculacarina* were 12, 3 and 4, respectively (S7 Table). Among the 3 haplotypes detected in *L. rufiantennus*, the one represented by 11 individuals was shared with *F. tonkinensis*, and the other two represented each by a single individual were private for *L. rufiantennus*. In the network from *COI* haplotypes (Fig 4B), haplotypes of *L. maculacarina* formed a separate clade, and those of *F. tonkinensis* and *L.*

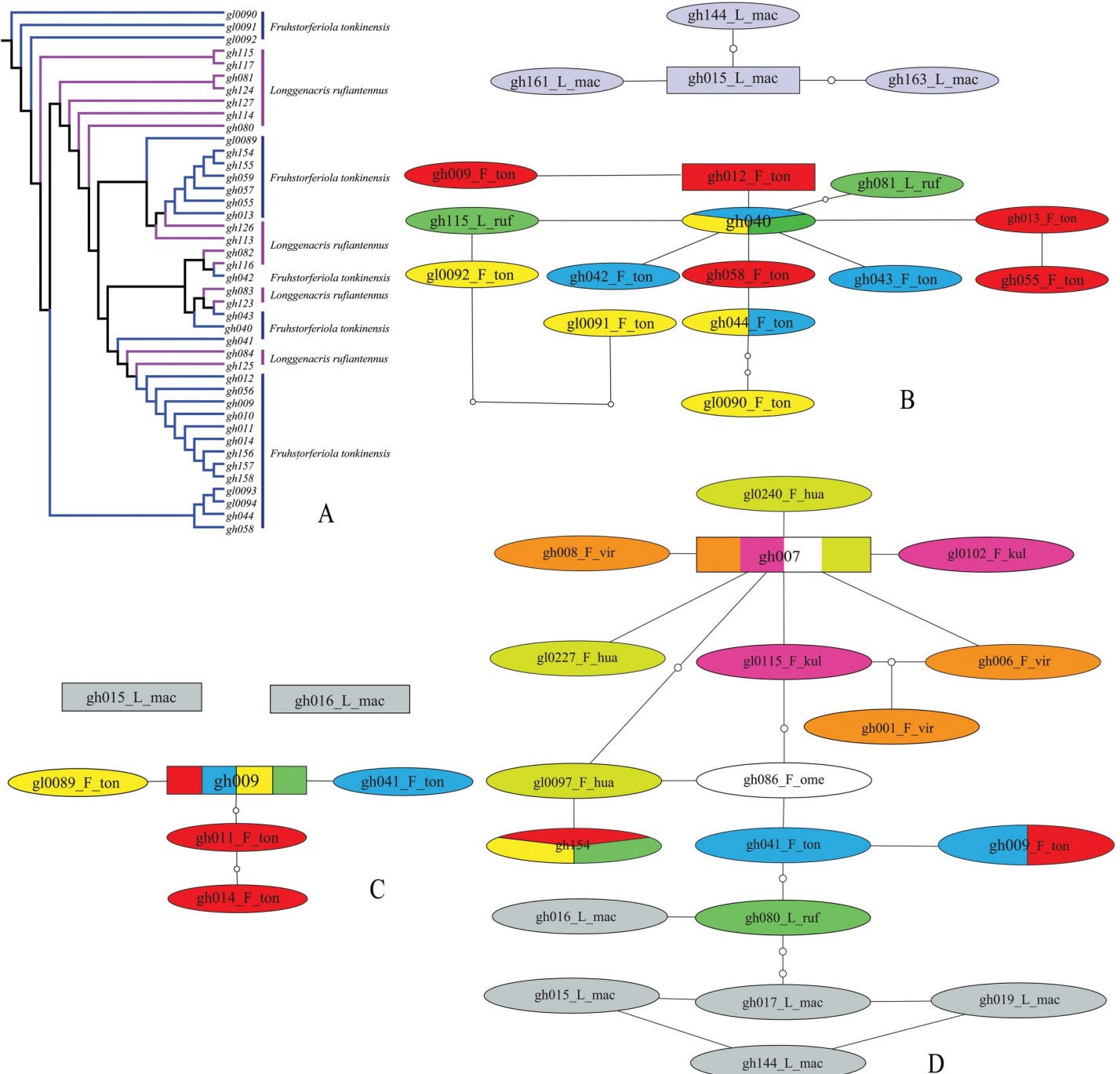

**Fig 4. NJ tree and haplotype networks of *F. tonkinensis+L. rufiantennus* group.** A. Subclade of NJ tree for *F. tonkinensis+L. rufiantennus* group reconstructed from *COI* gene. B. Haplotype network reconstructed from *COI* gene, C. Haplotype network reconstructed from *ITS1* sequence. D. Haplotype network reconstructed from *ITS2* sequence (including *F. viridifemorata* and *L. maculacarina*).

*rufiantennus* formed another clade together. In the clade of *F. tonkinensis+L. rufiantennus*, no haplotypes from the same population formed monophyletic subclade. For *ITS1* sequences, only 5 haplotypes were detected in *F. tonkinensis* and all individuals of *L. rufiantennus* shared the same haplotype with some individuals of *F. tonkinensis* from the 5 sampled populations

**Table 1. Intraspecific variation of and interspecific divergence between species of *F. tonkinensis*+*L. rufiantennus* group and *L. maculacarina* calculated from *COI* sequence.**

| Species | Intraspecific variation (Pairwise/mean) | Interspecific divergence (Pairwise/mean) | |
|---|---|---|---|
| | | *F. tonkinensis* | *L. rufiantennus* |
| *F. tonkinensis* | 0–1.08% (0.45%) | ---- | |
| *L. rufiantennus* | 0–0.46% (0.11%) | 0–0.77%(0.33%) | |
| *L. maculacarina* | 0–0.61% (0.17%) | 6.72–7.93% (7.40%) | 7.24–7.92% (7.33%) |

(S8 Table). In the network from *ITS1* sequences (Fig 4C), haplotypes of *F. tonkinensis* and *L. rufiantennus* formed a clade but no monophyletic subclade, and the 2 haplotypes of *L. maculacarina* did not connect into a single network, but separated from each other. For *ITS2* sequences, 3 haplotypes were detected for each of *F. tonkinensis* and *L. rufiantennus*, with 2 shared haplotypes (S9 Table). Haplotypes of all 3 species connected into a single network together with haplotypes of *F. viridifemorata* group (Fig 4D), indicating a much lower evolution rate in *ITS2* sequence.

In GMYC analysis based on *COI* sequences, 14 putative species were delineated from the whole data set (S10 Table, S4 Fig). The 10 individuals of *Paratonkinacris vittifemoralis* collected from the same locality (Gaozhai, Maoershan, Xing'an county, Guangxi) were delineated into 2 putative species, one represented by 9 individuals and the other by the single sample gl0251. Each population of *Emeiacris maculata* was delineated as an independent species. *F. tonkinensis* and *L. rufiantennus* were delineated as the same species (Fig 5C, S4 Fig; S10 Table). Samples of *F. viridifemorata* group were delineated into 4 putative species by neither

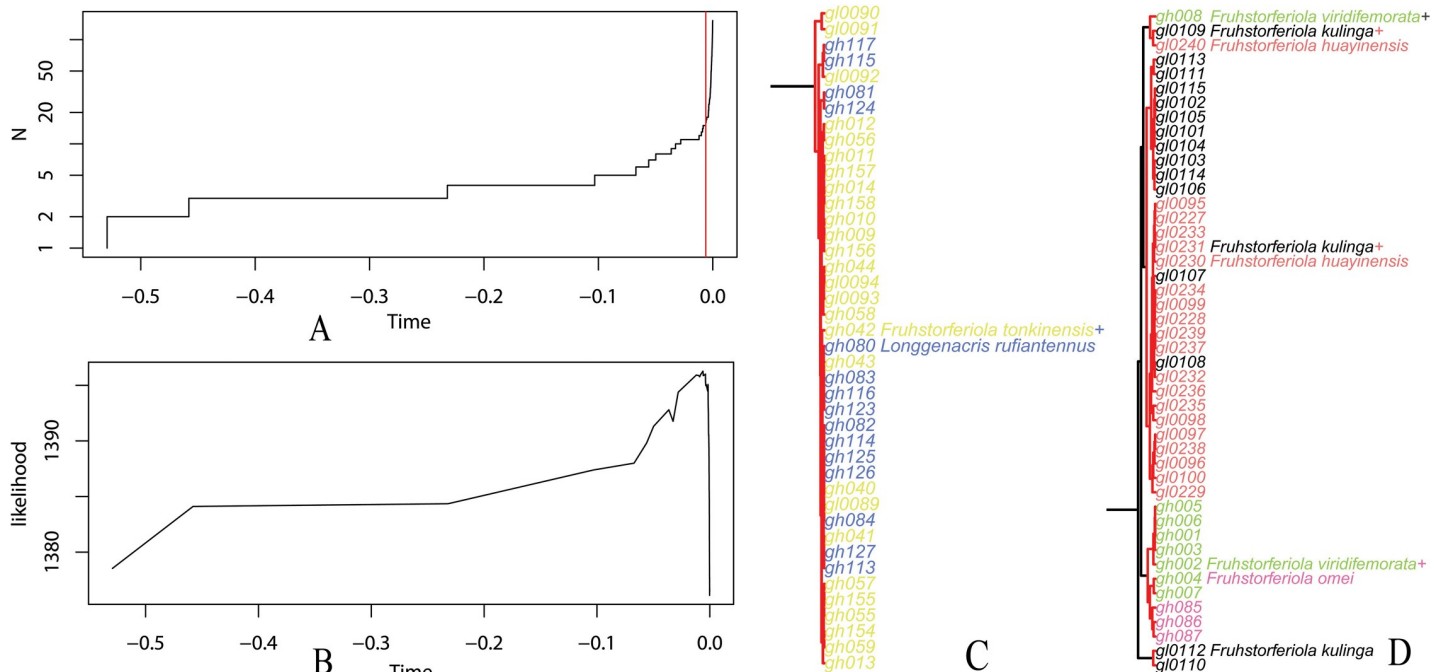

**Fig 5. Species delimitation according to the generalized mixed Yule coalescent (GMYC) single-threshold model using *COI* data set.** A. Lineage-through-time plot based on the ultrametric tree obtained from *COI* sequences. The sharp increase in branching rate, corresponding to the transition from interspecific to intraspecific branching events, is indicated by a red vertical line. The *x*- axes (both in panels A and B) show substitutions per nucleotide site. B. Likelihood function produced by GMYC to estimate the peak of transition between cladogenesis (interspecific diversification) and allele intraspecific coalescence along the branches. C. *F. tonkinensis*+*L. rufiantennus* subclade of the ultrametric tree. D. *F. viridifemorata* group subclade of the ultrametric tree.

morphospecies nor populations, and samples of each remaining species were delineated as an independent species.

**(2) *Fruhstorferiola viridifemorata* group.**   In an earlier study, the relationship between *F. kulinga* and *F. huayinensis* was discussed using single *COI* barcoding fragment, and the result did not support the validity of *F. huayinensis* [14]. Not only *F. kulinga* and *F. huayinensis* are difficult to distinguish morphologically, but also the other 5 species belonging to *F. viridifemorata* group display nearly no distinguishable morphological difference from each other. Therefore, we include *F. viridifemorata* and *F. omei* into the present analysis to explore the relationship among them again.

In the NJ tree of *COI* sequences, the four species of *F. viridifemorata* group formed a monophyletic clade. Although the 3 individuals of *F. omei* formed a so-called monophyletic subclade, but it completely fell into the larger subclade of *F. viridifemorata*. Individuals of the other 3 species clustered neither by species nor populations (Fig 6A). In the NJ tree of *ITS1* sequences, species of *F. viridifemorata* group did not form a monophyletic clade, but formed three separate clades and added in turn to the clade of *F. tonkinensis+L. rufiantennus* group together with one individual of *F. omei* and one of *F. viridifemorata* (S3B Fig). In the NJ tree of *ITS2* sequences, most individuals of *F. viridifemorata* group formed a monophyletic clade, but again clustered neither by species nor populations, with exceptions of 2 individuals, the one was gh086 of *F. omei* which clustered with a subclade of *O. longipennis+F. tonkinensis+L. rufiantennus*, and the other was individual gl0097 of *F. huayinensis* which clustered with the larger subclade of *O. longipennis+F. tonkinensis+L. rufiantennus*+gh086 (S3C Fig).

Mean intraspecific variations within each species calculated from *COI* sequences were distinctly less or slightly larger than 1%, and the largest pairwise intraspecific variation was as high as 2.97% in *F. kulinga*, but still slightly less than 3%. Broad overlaps between intraspecific genetic variations and interspecific divergences are found in all species pairs (Table 2). For *ITS1* and *ITS2* sequences, all intraspecific variations within population are distinctly less than 1% and only a few ones between populations are slightly more than 1% (S3 Table). As for the interspecific divergences, the genetic distances between species within the genus *Fruhstorferiola* were all less than 1%, and those between *Fruhstorferiola* species and the species in other genera were distinctly more than 2% (S5 and S6 Tables).

Haplotype network analysis detected no shared haplotype in *COI* sequences among the four species (S7 Table), but shared haplotypes occur in *ITS1* and *ITS2* sequences among these species (S8 and S9 Tables). In the network from *COI* haplotypes (Fig 6B), all haplotypes were connected into a large network in a maximum connection steps of 11 at 95%, but three of the four species did not form reciprocally monophyletic clades. Although the three haplotypes of *F. omei* formed a so-called monophyletic clades, the maximum mutational steps of haplotypes within *F. omei* reached 4 steps, slightly higher than the minimum mutational steps of haplotypes between *F. omei* and *F. viridifemorata*. For *ITS1* sequences, a haplotype shared by three species with high frequencies as well as another one shared by two species with low frequencies were found (S8 Table). In the network from *ITS1* haplotypes (Fig 6C), there was still no species forming reciprocally monophyletic clades. For *IIS2* sequences, a haplotype shared by four species was found (S9 Table) and all haplotypes of *F. viridifemorata* group and *F. tonkinensis+L. rufiantennus* group were connected into a single network as mentioned in the previous section (Fig 4D).

For the four putative species delineated in GMYC analysis (S10 Table), the putative species 9 consisted of two of the four individuals of *F. kulinga* from Longmenhe, Hubei Province, the putative species 10 consisted of all three individuals of *F. omei* from Emeishan, Sichuan Province and seven of the eight individuals of *F. viridifemorata* from Longwangshan, Zhejiang Province, the putative species 11 consisted of most individuals of *F. kulinga* and *F. huayinensis* from different localities, the putative species 12 consisted of one individual of each species of

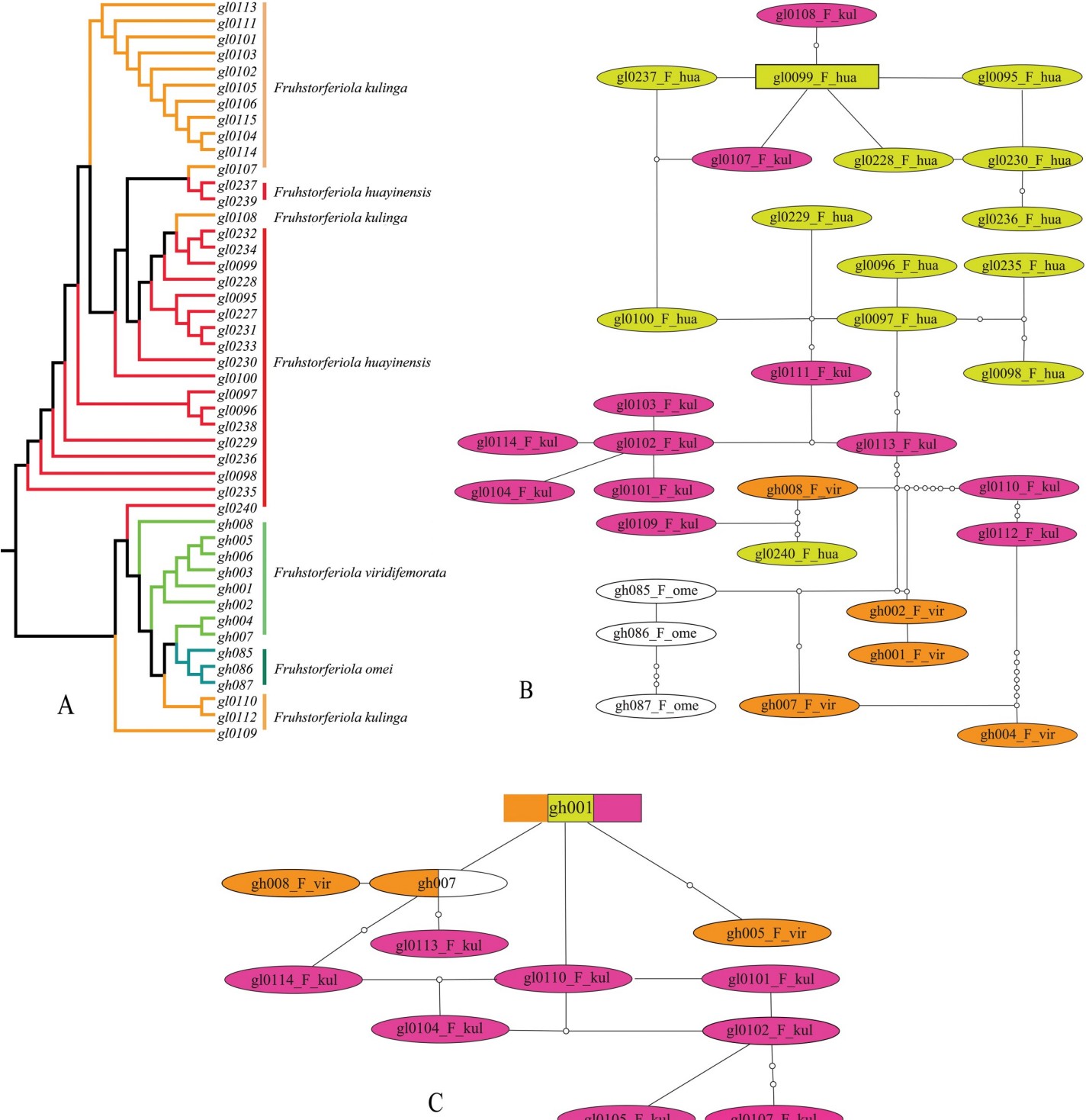

**Fig 6. NJ tree and haplotype networks of *F. viridifemorata* group.** A. Subclade of NJ tree for *F. viridifemorata* group reconstructed from *COI* gene. B. Haplotype network reconstructed from *COI* gene, C. Haplotype network reconstructed from *ITS1* sequence. D. Haplotype network reconstructed from *ITS2* sequence.

**Table 2. Intraspecific variation of and interspecific divergence between species of *F. viridifemorata* group calculated from *COI* sequence.**

| Species | Intraspecific variation (Pairwise/mean) | Interspecific divergence (Pairwise/mean) | | |
|---|---|---|---|---|
| | | *F. viridifemorata* | *F. omei* | *F. huayinensis* |
| *F. viridifemorata* | 0–1.23% (0.60%) | - - - - | - - - - | - - - - |
| *F. omei* | 0.15–0.77% (0.51%) | 0.46–1.54% (1.00%) | - - - - | - - - - |
| *F. huayinensis* | 0–1.85% (0.57%) | 0.61–2.48% (1.91%) | 1.23–2.80% (2.03%) | - - - - |
| *F. kulinga* | 0–2.97% (1.02%) | 0.46–2.32% (1.48%) | 0.92–2.64% (1.65%) | 0.15–2.96% (1.12%) |

*F. kulinga* and *F. huayinensis* and *F. viridifemorata*. Three of the four putative species, each consisting of individuals from vast area, contained individuals of at least two morphospecies, and individuals in three populations (Baiyunshan, Longmenhe, Longwangshan) were assigned to at least 2 GMYC species.

## Discussion

### Phylogenetic position and species delimitation of *Longgenacris rufiantennus*

Although being placed in the genus *Longgenacris* originally, *L. rufiantennus* has substantial differences from its congener *L. maculacarina* concerning the length of tegmina and wings, the shape of cerci in male, the subgenital plate in female as well as the structure of male genitalia, and shows no morphological difference from *F. tonkinensis* [16]. Phylogeny reconstructed from different datasets consistently supported the closer relationship of *L. rufiantennus* with *F. tonkinensis*, and *L. maculacarina* usually formed an independent monophyletic clade as a sister group of the genus *Fruhstorferiola* (Fig 3A and 3D, S1 and S2 Figs). Therefore, *L. rufiantennus* should be regarded as a member of the genus *Fruhstorferiola* but not a member of *Longgenacris* no matter according to morphological or molecular evidences.

As for the relationship between *L. rufiantennus* and *F. tonkinensis*, all analysis (NJ tree, genetic distance and haplotype network) led to the same result that they should be the same species but not two independent species because all individuals of *L. rufiantennus* fall into the clade of *F. tonkinensis* in NJ trees (Fig 4A, S3 Fig), the pairwise genetic distances within *F. tonkinensis* completely overlapped with those between *F. tonkinensis* and *L. rufiantennus* (Table 1), the *COI* haplotype of *L. rufiantennus* with highest frequency were shared with *F. tonkinensis* (S7 Table) and all haplotypes of the two species formed a whole network under the 95% parsimony connection limit (Fig 4B), GMYC analysis delineated them as the same species (Fig 5C, S4 Fig; S9 Table). Therefore, this study confirmed the synonymy of *L. rufiantennus* with *F. tonkinensis* [16].

### Subfamily placement of the genus *Longgenacris*

The genus *Longgenacris* was originally placed in the subfamily Melanoplinae and considered most similar to the genus *Ognevia* Ikonnikov, 1911 [15]. The phylogenetic position of the genus was discussed recently based on morphological characters because once it was regarded as a member of the subfamily Catantopinae [16]. In this study, the genus *Longgenacris* consistently has the most close relationship with and is most of the time the sister group of the genus *Fruhstorferiola* (Fig 3A and 3D, S1–S3 Figs). Therefore, this study supports the original placement of the genus *Longgenacris* in the subfamily Melanoplinae.

### Species delimitation of *Fruhstorferiola viridifemorata* group

To explore the species boundary among species in *F. viridifemorata* group in a larger scale than the previous study [14], samples of two additional species, i.e. *F. viridifemorata* and *F.*

*omei*, were added to the present study, and *ITS* region was employed in addition to *COI* sequence. However, the increases of the sampled species and molecular markers did not lead to different result from that of previous study [14]. It seemed that the resolution of the datasets were contributed mainly by *COI* gene sequences, and *ITS* region had a much lower evolution rate than *COI* gene in our datasets. No matter the non-monophyly of the morphospecies in NJ trees, the extent of the overlaps between pairwise intraspecific genetic variations and interspecific divergences, or the haplotype networks, all results did not support the validity of the four independent morphospecies, and this was consistent with the results of our morphological recomparison mentioned in introduction section. As for the result of GMYC analysis, we will discuss it in detail in the following section.

### Cryptic species or genetic polymorphism: testing species hypotheses with diagnostic characters from different approaches

In the case of *L. rufiantennus*, a comprehensive comparison across members of closely related genera revealed high morphological similarity between *L. rufiantennus* and *F. tonkinensis*, and a synonymy was proposed based on morphological evidences [16]. This decision is confirmed by molecular evidences again in this study, resulting in a perfect synergy of resolution that an integrated taxonomy is capable of attaining [2].

In the case of *F. viridifemorata* group, the condition is a little more complicated. Although NJ tree, pairwise genetic distances and haplotype networks retrieved coincident results corresponding to the result of morphological recomparison, the GMYC analysis of *COI* gene delineated four molecular operational taxonomic units (MOTUs) from samples of *F. viridifemorata* group (Fig 5D, S4 Fig; S10 Table). Do the four MOTUs represent morphologically cryptic species or only ancient genetic polymorphism? Among species of *F. viridifemorata* group, the morphological characters originally employed to describe the different species have been approved to be variable even within populations of the same species (Fig 2), and most analyses of molecular evidences are congruent with the result of morphological reexamination. As for the four MOUTs delineated by GMYC analysis using *COI* gene (S10 Table), they didn't be supported by either morphological or geographical informations. Furthermore, this approach tends to overestimate the number of species because of errors in reconstruction of ultrametric input trees [49, 50], or in the presence of high population structure or considerably high values of effective population size [51, 52], especially when mitochondrial genomic dataset is employed [11]. Although GMYC was considered a robust tool for delimiting species when only single-locus information was available [53], it cannot be used as sufficient evidence for evaluating the specific status of particular cases without additional data [54]. Therefore, we can't be able to break out of the taxonomic circle at present, and prefer to consider the four MOTUs of *F. viridifemorata* group delineated with GMYC model as ancient genetic polymorphism. The diverse and complicated relationships of Longmenhe population of *F. kulinga* with other species (Fig 3C and 3F) indicate the possibility that Longmenhe population has the highest genetic diversity and might be a centre of dispersal for a widespread species. This molecular study will serve as a robust basis to carry out further studies using additional molecular markers and morphological informations from different character systems.

Although we increased the numbers of sampled species and molecular markers in this study, a sample size of three individuals for *F. omei* was a little insufficient, no individual from type localities was sampled for *F. viridifemorata* and *F. kulinga*, and molecular markers employed were still not enough. Considering the genomic features of species complex in early stage of parallel speciation or divergence where conflicting inferences are more prone to appear [7, 13], the discordant pattern between mitochondrial and nuclear DNA [19, 55, 56],

and the possibility of the concurrence of both cryptic species and morphological polymorphism in the same group [57], a more comprehensive study combining complete mitochondrial genome, more nuclear genes and morphological data is going to carry out. Anyway, the consensus of numerous independent criteria is needed to define species boundaries, particularly in cases of recent speciation events or species that are very similar and difficult to distinguish morphologically [9, 58]. We believe that a more unambiguous outline of the relationship within *F. viridifemorata* group will be achieved with the accumulation of more types of informations.

## Supporting information

**S1 Table. Materials involved in this study.**
(DOCX)

**S2 Table. Mapping table between GenBank accession numbers and voucher numbers.**
(DOCX)

**S3 Table. Intraspecific variations calculated from different datasets.**
(DOCX)

**S4 Table. Mean genetic distances between species calculated from *COI* alignment.**
(DOCX)

**S5 Table. Mean genetic distances between species calculated from *ITS1* alignment.**
(DOCX)

**S6 Table. Mean genetic distances between species calculated from *ITS2* alignment.**
(DOCX)

**S7 Table. Haplotyptes of *COI* detected from samples of *F. viridifemorata* and *F. tontinensis* +*L. rufiantennus* groups.**
(DOCX)

**S8 Table. Haplotypes of *ITS1* detected from samples of *F. viridifemorata* and *F. tontinensis*+*L. rufiantennus* groups.**
(DOCX)

**S9 Table. Haplotypes of *ITS2* detected from samples of *F. viridifemorata* and *F. tontinensis*+*L. rufiantennus* groups.**
(DOCX)

**S10 Table. Putative species delineated from *COI* alignment using GMYC model.**
(DOCX)

**S1 Fig. Phylogeny deduced in maximum likelihood framework from alignment of *ITS1* sequences.**
(DOCX)

**S2 Fig. Phylogeny deduced in maximum likelihood framework from alignment of *ITS2* sequences.**
(DOCX)

**S3 Fig. NJ trees reconstructed from single and combined alignments of *COI*, *ITS1* and *ITS2*.**
(DOCX)

**S4 Fig. Species delimitation according to the generalized mixed Yule coalescent (GMYC) single-threshold model using *COI* dataset.**
(DOCX)

## Acknowledgments

We would like to thank Mr. Tao Wang for his help in implementing GMYC analysis, to thank Dr. Xiaolong Lin for his comments on species delimitation methods. . ..

## Author Contributions

**Conceptualization:** Yuan Huang, Jianhua Huang.

**Data curation:** Jianhua Huang.

**Formal analysis:** Jingxiao Gu, Bing Jiang, Haojie Wang, Jianhua Huang.

**Funding acquisition:** Liliang Lin, Jianhua Huang.

**Investigation:** Tao Wei, Liliang Lin.

**Supervision:** Yuan Huang, Jianhua Huang.

**Visualization:** Jingxiao Gu, Bing Jiang.

**Writing – original draft:** Jingxiao Gu, Bing Jiang.

**Writing – review & editing:** Jianhua Huang.

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
