## [Decision Letter · Decision Letter 0]

11 Jun 2020

PONE-D-20-12129

Phylogeny and species delimitation of the genus Longgenacris and Fruhstorferiola viridifemorata species group (Orthoptera: Acrididae: Melanoplinae) based on molecular evidence

PLOS ONE

Dear Dr. Huang,

Thank you for submitting your manuscript to PLOS ONE. After careful consideration, we feel that it has merit but does not fully meet PLOS ONE’s publication criteria as it currently stands. Therefore, we invite you to submit a revised version of the manuscript that addresses the points raised during the review process.

We look forward to receiving your revised manuscript.

Kind regards,

Feng ZHANG, Ph.D.

Academic Editor

PLOS ONE

Journal Requirements:

Reviewers' comments:

Reviewer's Responses to Questions

**Comments to the Author**

1. Is the manuscript technically sound, and do the data support the conclusions?

Reviewer #1: Yes

Reviewer #2: Yes

2. Has the statistical analysis been performed appropriately and rigorously? 

Reviewer #1: Yes

Reviewer #2: Yes

3. Have the authors made all data underlying the findings in their manuscript fully available?

Reviewer #1: Yes

Reviewer #2: Yes

4. Is the manuscript presented in an intelligible fashion and written in standard English?

Reviewer #1: Yes

Reviewer #2: No

5. Review Comments to the Author

Reviewer #1: The content of this manuscript is quite good. For phylogeny reconstruction, I don't think use Tetrigidae and Tettigoniidae as outgroups is right choice. Furthermore, I hope the author adds Bayesian inference analysis. The quality of English needs improving.

Other minor points are:

line 36 Delete Keyword “Longgenacris" or "Longgenacris rufiantennus";

line 104 "cytochrome c oxidase subunit I (COI, cox1)"→"COI or cox1";

line 122 "100% ethanol"→"anhydrous ethanol";

line 140-141 "PCR products were sent to the biological company", Which company？

line 183 "TSC1.21"→"TCS1.21";

Reviewer #2: This article deserves publication. However, there are the following problems.

1) Normally, as for the raltionship between species and above, NJ tree is not optimal selection.

2) On the page 16, line 403, “ITS region had a much lower evolution rate than COI gene in our datasets”, I’m very interested in it. Can the author explain this?

3) All figures are not clear. Low photo resolution.

4) English speakers are requested to polish this draft.

6. PLOS authors have the option to publish the peer review history of their article (what does this mean?). If published, this will include your full peer review and any attached files.

Reviewer #1: No

Reviewer #2: No

---

## [Author Response · Author response to Decision Letter 0]

1 Aug 2020

1. Adds Bayesian inference analysis. We have reconstructed BI tree using different datasets, but the results is similar to ML tree and have no significant difference. Therefore, we think there is no need to add the figures of BI trees to the manuscript. The draft of the BI tree has been inserted into the rebuttal letter.

2. Using Tetrigidae and Tettigoniidae as outgroups is not a right choice. Outgroup(s) is only a reference group to improve further the acuracy of the topology inferred. So, a group of any taxa should be appropriate outgroup only if it does not decrease the quality of the tree topology. In our study, selecting Tetrigidae and Tettigoniidae as outgroups is feasible because no such problem occurred in our study. 

3. Using Tetrigidae and Tettigoniidae as outgroups is not a right choice. Outgroup(s) is only a reference group to improve further the acuracy of the topology inferred. So, a group of any taxa should be appropriate outgroup only if it does not decrease the quality of the tree topology. In our study, selecting Tetrigidae and Tettigoniidae as outgroups is feasible because no such problem occurred in our study. 

4. On the page 16, line 403, “ITS region had a much lower evolution rate than COI gene in our datasets”, I’m very interested in it. Can the author explain this? At the moment, we cannot be able to explain such a special case, but we will follow with interest in the future.

---

## [Editor Report · Decision Letter 1]

5 Aug 2020

Phylogeny and species delimitation of the genus Longgenacris and Fruhstorferiola viridifemorata species group (Orthoptera: Acrididae: Melanoplinae) based on molecular evidence

PONE-D-20-12129R1

Dear Dr. Huang,

We’re pleased to inform you that your manuscript has been judged scientifically suitable for publication and will be formally accepted for publication once it meets all outstanding technical requirements.

Kind regards,

Feng ZHANG, Ph.D.

Academic Editor

PLOS ONE
---

## [Editor Report · Acceptance letter]

17 Aug 2020

PONE-D-20-12129R1 

Phylogeny and species delimitation of the genus Longgenacris and Fruhstorferiola viridifemorata species group (Orthoptera: Acrididae: Melanoplinae) based on molecular evidence 

Dear Dr. Huang:

I'm pleased to inform you that your manuscript has been deemed suitable for publication in PLOS ONE. Congratulations! Your manuscript is now with our production department. 

Kind regards, 

on behalf of

Dr. Feng ZHANG 

Academic Editor

PLOS ONE